# Direct modulation of aberrant brain network connectivity through real-time NeuroFeedback

**Michal Ramot[1]\*, Sara Kimmich[1], Javier Gonzalez-Castillo[1], Vinai Roopchansingh[2], Haroon Popal[1], Emily White[1], Stephen J Gotts[1], Alex Martin[1]**

[1]Laboratory of Brain and Cognition, National Institute of Mental Health, National Institutes of Health, Bethesda, United States; [2]Functional MRI Facility, National Institute of Mental Health, National Institutes of Health, Bethesda, United States

**Abstract** The existence of abnormal connectivity patterns between resting state networks in neuropsychiatric disorders, including Autism Spectrum Disorder (ASD), has been well established. Traditional treatment methods in ASD are limited, and do not address the aberrant network structure. Using real-time fMRI neurofeedback, we directly trained three brain nodes in participants with ASD, in which the aberrant connectivity has been shown to correlate with symptom severity. Desired network connectivity patterns were reinforced in real-time, without participants' awareness of the training taking place. This training regimen produced large, significant long-term changes in correlations at the network level, and whole brain analysis revealed that the greatest changes were focused on the areas being trained. These changes were not found in the control group. Moreover, changes in ASD resting state connectivity following the training were correlated to changes in behavior, suggesting that neurofeedback can be used to directly alter complex, clinically relevant network connectivity patterns.

DOI: https://doi.org/10.7554/eLife.28974.001

**\*For correspondence:**
michal.ramot@nih.gov

**Competing interests:** The authors declare that no competing interests exist.

## Introduction

Autism Spectrum Disorder (ASD) refers to a group of neurobiological disorders, which affect a growing proportion of the population. Patients with ASD suffer from a range of social and communication impairments, along with other characteristic behaviours and deficits. Behavioural treatment options are limited in their efficacy, and often do not generalize well beyond the specific training paradigm (*Otero et al., 2015*; *Williams White et al., 2007*). Numerous studies have documented widespread patterns of aberrant brain functional connectivity in patients with ASD, involving many cortical regions including frontal, parietal, and temporal lobes (*Müller et al., 2011*; *Picci et al., 2016*; *Di Martino et al., 2014*). Specifically, these studies show that multiple cortical areas are significantly under-connected in patients with ASD compared to typically developing (TD) control subjects, although over-connectivity has also been reported (*Hahamy et al., 2015*; *Belmonte et al., 2004*). At the individual level, the degree of diminished connectivity as well as measures of cortical thickness were found to be correlated with symptom severity (*Gotts et al., 2012*; *Wallace et al., 2012*; *Di Martino et al., 2009b*; *Assaf et al., 2010*) and this measure of connectivity was even predictive of future progression of autistic symptoms (*Plitt et al., 2015*). Together, these findings suggest a causal link between connectivity and behavior, such that changing the connectivity might lead to a change in behavior. Traditional, explicit training paradigms, activate these aberrant networks (*Kana et al., 2009*; *Di Martino et al., 2009a*), thus potentially reinforcing the sub-optimal connectivity. An implicit training paradigm, which allows the direct reinforcement of the desired networks,

**eLife digest** Even when we are at rest, our brains are always active. For example, areas of the brain involved in vision remain active in complete darkness. Different brain regions that connect together to perform a given task often show coordinated activity at rest. Past studies have shown that these resting connections are different in people with conditions such as autism. Some brain regions are more weakly connected while others are more strongly connected in people with autism spectrum disorder compared to those without. Furthermore, people with more severe symptoms seem to have more abnormal connections.

"Neurofeedback training" is a method of changing the resting connections between different brain regions. Scientists measure a brain signal – the connection between different brain regions – from a person in real time. They then provide positive feedback to the person if this signal improves. For example, if a connection that is too weak becomes stronger, the scientists might reinforce this by providing feedback on the success. Previous work has shown that neurofeedback training may even change people's behaviour. However, it has not yet been explored as a method of treating the abnormal connections seen in people with autism when their brains are at rest.

To address this, Ramot et al. used a technique known as "functional magnetic resonance imaging" (or fMRI for short) to measure brain activity in young men with autism. First, certain brain regions were identified as having abnormal resting connections with each other. The participants were then asked to look at a blank screen and to try to reveal a picture hidden underneath. Whenever the connections between the chosen brain regions improved, part of the picture was revealed on the screen, accompanied by an upbeat sound. The participants were unaware that it was their brain signals causing this positive feedback.

This form of neurofeedback training successfully changed the abnormal brain connections in most of the participants with autism, making their connections more similar to those seen in the wider population. These effects lasted up to a year after training. Early results also suggest that these changes were related to improvements in symptoms, although further work is needed to see if doctors could reliably use this method as a therapy. These findings show that neurofeedback training could potentially help treat not only autism spectrum disorder, but a range of other disorders that involve abnormal brain connections, including depression and schizophrenia.

DOI: https://doi.org/10.7554/eLife.28974.002

bypassing the atypical activation induced by explicit tasks, might be a better candidate for potential intervention in ASD.

Real-time fMRI neurofeedback (rt-fMRI-nf) is an emerging technique with great potential for clinical applications (*Stoeckel et al., 2014*; *Sulzer et al., 2013*; *Weiskopf, 2012*; *Birbaumer et al., 2013*). With this technique, network states can be monitored in real-time, and desired states can be reinforced through positive feedback. Covert neurofeedback is a variant of neurofeedback, in which participants are given no strategy with which to control the feedback, and might not even be cognizant that the feedback is related to brain activity. This tool is therefore extremely flexible, as it does not require the formulation of a specific strategy and is not limited by what we know about the networks or the ways in which they are typically activated. Instead, desired states are reinforced when they occur spontaneously, allowing for implicit training of networks, such as those found to be under-connected in ASD. We designed a covert neurofeedback experiment, to test whether it would be possible to change connectivity between these aberrantly connected network nodes, through direct reinforcement of spontaneously occurring network states. This decision is motivated by recent work showing that positive and negative reinforcement of brain activity patterns are sufficient for promoting small but widespread changes in network connectivity, even without any learning intention on the part of participants (*Ramot et al., 2016*).

Further evidence that covert neurofeedback can change networks, and that these changes can have behavioral effects, comes from other recent work. A number of studies have been successful at training complex patterns of activity within a given network using multi-voxel pattern analysis (MVPA) techniques (*deBettencourt et al., 2015*; *Amano et al., 2016*; *Shibata et al., 2011*), with feedback related changes corresponding to robust behavioural changes after only a few sessions,

and significant change being detectable after as little as one training session. Feedback induced behavioural changes have been shown to range from very local and low level, such as changes in perception of line orientation after V1 training (*Shibata et al., 2011*), or inducing color association in V1 in a somewhat less local paradigm (*Amano et al., 2016*), to changes in high level functions such as attention (*deBettencourt et al., 2015*), and fear perception (*Koizumi et al., 2016*). Such changes can even be bi-directional, both behaviourally and at the network level (*Cortese et al., 2017*). This previous work sets up covert neurofeedback as a good candidate for a potential intervention in ASD, though whether specific, long-ranging connectivity changes can be induced through neuro-feedback, which regions/networks are amenable to such reinforcement, how such training will affect wider brain networks, and how long these changes last, are all still open questions. In this study, we lay out a proof of principle of the plausibility of such training, showing robust, long-lasting feedback induced changes in these aberrant networks, coupled with preliminary results as to the behavioural correlates of these changes. 17 ASD participants and 10 control participants were scanned over multiple sessions (123 sessions in total). These results reflect not only on the potential uses of such training in ASD, but also in other disorders with underlying aberrant connectivity at their core.

## Results

### Selection of training targets

We used previously collected resting state data on large groups of ASD and TD participants (N = 56 ASD, 62 TD) to identify two target brain regions that showed large under-connectivity in ASD compared with TD individuals, while also being physically distant from each other, and belonging to separate networks (*Figure 1*): target1 in superior temporal sulcus (STS) and target2 in somatosensory cortex, both of which have been consistently implicated in social processing (*Allison et al., 2000*; *Frith and Frith, 2010*; *Adolphs, 2009*; *Damasio et al., 2000*), and have previously been found to be under-connected and atypically activated in ASD (*Chen et al., 2015*; *Gotts et al., 2012*; *Müller et al., 2001*; *Tuttle et al., 2016*; *Khan et al., 2015*; *Khan et al., 2013*). This dataset is an expansion of previously reported data (*Gotts et al., 2012*), which found very similar aberrant

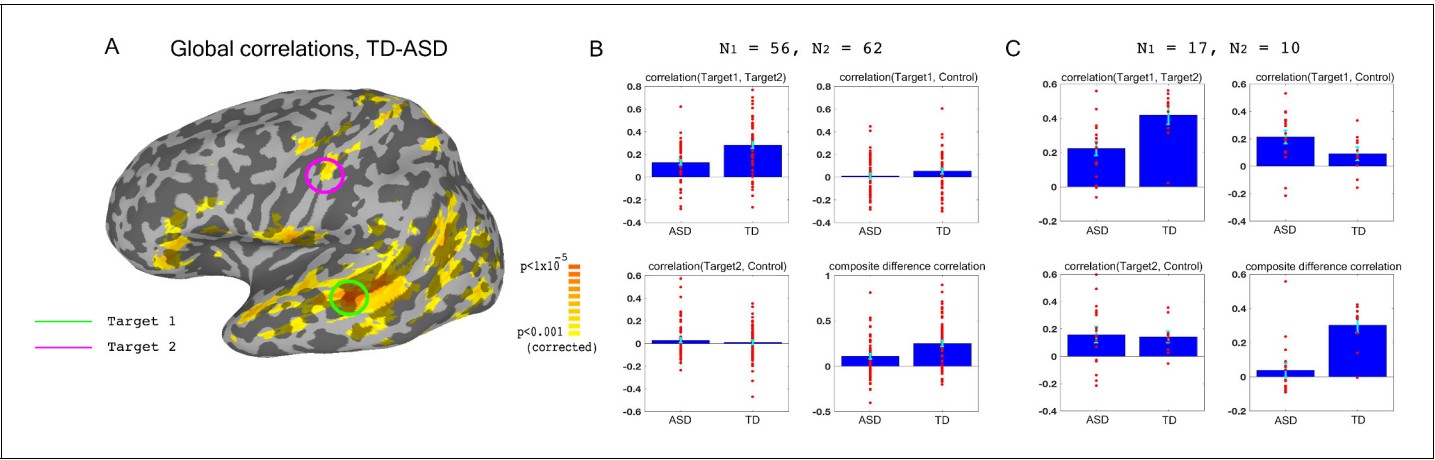

**Figure 1.** Choosing ROIs. (A) Group differences between TD (N = 62) and ASD participants (N = 56), matched for motion, age and IQ. Difference maps calculated on the average correlation of each voxel with all other grey matter voxels in the brain. Target1 was chosen as the region with the greatest between-group difference, and Target2 was chosen as the region in which the difference in connectivity to Target1 was greatest between groups, while also being in a physically distant, distinct network based on (*Gotts et al., 2012*). (B) Pairwise correlations for the dataset shown in (A), between the two targets (top left), target1 and control (top right), target2 and control (bottom left), and the composite difference measure, based on the difference in correlations between the two targets and the target-control pairs (see Materials and methods). Difference between the ASD and TD groups is significant for target1-target2 correlations (p=4.3×10⁻⁵), and the composite difference measure (p=0.001). (C) Same as (B), but for the current cohort of participants who took part in the neurofeedback study. Correlations are averaged across the first two rest scans of the first day, before training. Between group difference is significant for target1-target2 correlations (p=0.002), target1-control correlations (p=0.04), and the composite difference measure (p<1×10⁻⁴). Blue bars represent the subject mean, cyan error bars mark ± SEM. Red dots represent each individual subject.
DOI: https://doi.org/10.7554/eLife.28974.003

connectivity patterns, matching results from other studies using large datasets (*Cheng et al., 2015*). As in the previously published subset of the dataset, we found that under-connectivity between these two networks (STS and somatosensory) in ASD was significant in both this large dataset ($p<4.3\times10^{-5}$) and in the 17 participants recruited for the neurofeedback study (p=0.002), as well as significantly correlated to social symptom severity, as measured by the Social Responsiveness Scale (SRS) (r = $-0.35$ p<0.009 without regressors, r = $-0.31$, p<0.026, using age and motion as regressors, r values represent Pearson's correlation, p-values determined by permutation tests). The SRS is a parent filled questionnaire, which is designed to be a continuous, cardinal measure of social symptom severity in ASD, and has been shown to correlate with functional brain connectivity measures in multiple studies (*Anderson et al., 2011*; *Di Martino et al., 2009b*). This result indicates that connectivity between these two networks is clinically relevant, i.e. the lower the connectivity, the more severe the social symptoms (higher score on the SRS). The first goal of the training was therefore to increase the connectivity between target1 in STS, and target2 in somatosensory cortex.

In order to ascertain that we would only be reinforcing connectivity between our two targets, rather than global changes that cause an overall increase in correlations across the entire brain in an undifferentiated manner, we selected a third control region (in the inferior parietal lobule or IPL, part of the default mode network), which was chosen for being uncorrelated to the two target regions in our dataset of TD participants during resting state. IPL was significantly over-correlated to STS target1 in the ASD cohort participating in this study (*Figure 1C*). This combination of under-connectivity between STS and somatosensory with over-connectivity to the default mode network, is in line with recent evidence of reduced within-network cohesion coupled with reduced between-network differentiation (*Hahamy et al., 2015*; *Keown et al., 2016*). The goal of the neurofeedback training was therefore to induce greater differentiation between these three regions of interest (ROIs) in participants with ASD, so as to bring connectivity levels between those three networks closer to those of TD individuals. This meant increasing connectivity between the two target regions, while simultaneously decoupling the two target-control pairs. To this end, we came up with the composite difference measure, combining the target-target and target-control correlations (see Materials and methods). This measure was also significantly different between the ASD group and the TD group in both the previous large dataset (p=0.001), and in our cohort ($p<1\times10^{-4}$), *Figure 1B–C*). All p values were calculated through permutation tests, maintaining the original number of participants for each group.

## Training paradigm

For the initial part of the study, 17 patients with ASD participated in four training sessions, over the course of 8 days (two sessions of two consecutive days each, a week apart). Each session consisted of two rest scans, followed by four neurofeedback training scans, and finally two more rest scans (Each scan was 9 min in duration. See *Figure 2*). During the neurofeedback scans, participants started with a blank screen, and were instructed to attempt to reveal the picture hidden underneath (see *Figure 2—figure supplement 1* for an example). This was described to them as a puzzle task. No further instructions were given. Parents filled out behavioral questionnaires before training began, and two weeks after the last training session. An additional follow up study was then carried out, in which 15 of the 17 original participants returned for a final, slightly shorter training session. The interval between the original training and the follow up varied greatly between subjects, and ranged from 5 to 56 weeks.

One of the barriers to carrying out connectivity-based rt-fMRI-nf has been the slow timescale of fMRI recordings, making online calculations of correlations very limited. We therefore developed a method that can approximate the correlations using only two time points: every two seconds, for each TR (time to repetition), the signals from the three ROIs were analyzed in real-time (see Materials and methods), and the trend in the signal compared to the previous TR was noted for each of the three ROIs (increase/decrease). Positive feedback, in the form of revealing a part of the picture accompanied by an upbeat sound, was given whenever the network was deemed to have reached its desired state. As our goal was to increase correlation between the two target regions, and decrease correlations between the target and the control regions, feedback (i.e. revealing a part of the picture) was given whenever the signal trend in the two target ROIs was the same, and opposite from the trend in the control ROI (*Figure 2*, Materials and methods). This 'two-point' method was validated as being a good proxy for correlation analysis by comparing the results from this to

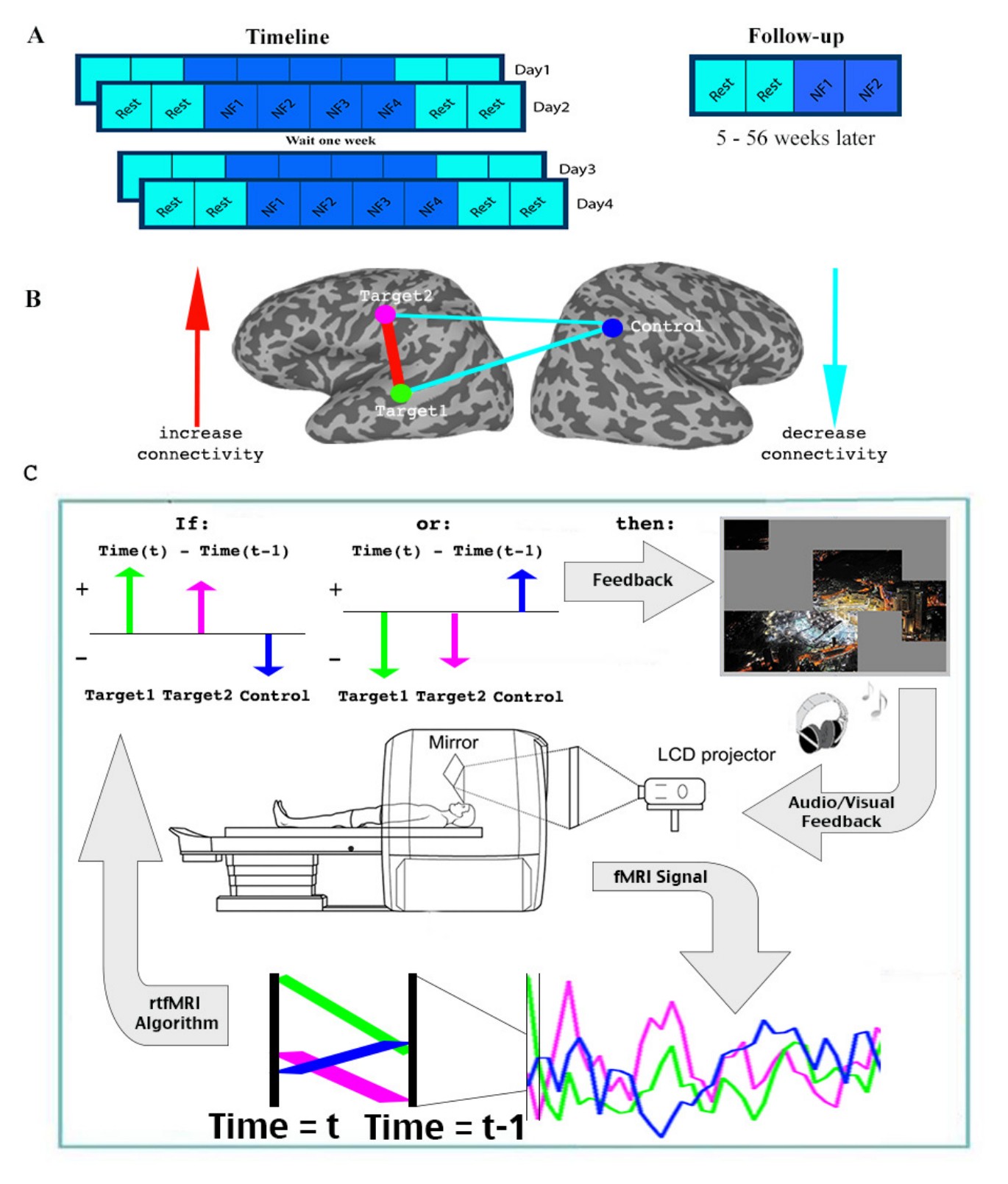

**Figure 2.** Experimental paradigm. (A) Timeline. (B) Location of ROIs, and network being trained. (C) Feedback session. Data was collected and analyzed in real-time, and a decision whether to present feedback (reveal a square of the picture +positive sound) was made based on the change in signal from one time (t-1) to the next (t) in the three ROIs. Feedback was given if the direction of change in the two targets was the same, and opposite from the direction of change in the control ROI. See.

*Figure 2 continued on next page*

*Figure 2 continued*

DOI: https://doi.org/10.7554/eLife.28974.004

The following figure supplements are available for figure 2:

**Figure supplement 1.** Puzzle task examples.

DOI: https://doi.org/10.7554/eLife.28974.005

**Figure supplement 2.** Validating the feedback method.

DOI: https://doi.org/10.7554/eLife.28974.006

**Figure supplement 3.** TD control group training.

DOI: https://doi.org/10.7554/eLife.28974.007

standard Pearson's correlation offline (r = 0.61, $p<1\times10^{-4}$ permutation test, *Figure 2—figure supplement 2*). We carried out simulations using more time points (2–6 TRs) to evaluate if longer time scales would lead to connectivity estimates for neurofeedback that better resemble actual correlations computed offline. We did not find any significant increase in the correlation of these extended measures to the whole series Pearson's correlations, which were our 'gold standard'. We therefore chose to provide feedback on just two time points, to minimize the timescale of the feedback.

At the end of each neurofeedback scan, participants were presented with their score, i.e. how many picture squares they had managed to reveal. They were then given a chance to attempt to beat their score on the next run, to win an additional bonus on top of the normal study compensation. The pictures were chosen to be neutral, depicting mostly scenes devoid of people and text, or abstract art/objects. Random pictures were chosen for each run, from a large set of such pictures. Participants completed 2–3 puzzles per scan on average, and there was no significant change in number of puzzles completed between day1 and day4.

Participants were blind to the purpose of the study, as well as to the mechanism of the neurofeedback, and even to the fact that it was neurofeedback. This was ascertained by exit questionnaires at the end of the last day of scanning, in which participants were interviewed regarding their thoughts on the study, their motivation, and their strategies during the training (see *Table 1* for responses). Responses as to the perceived nature of the 'puzzle task' varied widely, as did reported strategies, but none held any resemblance to the neurofeedback algorithm. Strategies mostly revolved around different ways of looking at the picture, as it was being revealed. Despite not knowing what they were supposed to do, most participants were highly motivated to solve the puzzles, with only 3 of the 11 participants for which responses are available reporting a motivation score of less than 5 (on a scale of 1–10, see *Table 1*).

## Control group

An additional control group of 10 TD participants completed the same initial 4 day training regimen, following the same protocol as the above. This group received feedback on the same three nodes, but in a different network configuration: target1 in the STS remained the same, but somatosensory target2 and the IPL control region switched roles, so that feedback was given whenever STS (target1) and IPL (now target2) were co-modulated, and were opposite to somatosensory cortex (now control), see *Figure 2—figure supplement 3*. This provided feedback orthogonal to that given to the ASD group. Another key difference is that this feedback was antithetical to the normal connectivity patterns found in the typically developing brain, as STS and somatosensory are well correlated in the typically developing brain, whereas the IPL region used in this study was explicitly chosen to be as uncorrelated as possible with STS in TDs during rest (*Figure 1B–C*). This control therefore served a dual purpose: in terms of the network that the ASD participants were being trained on, which rewarded increased connectivity between STS and somatosensory and decoupling of these from IPL, this was random feedback. That is to say, the feedback given to the TD participants was uncorrelated with the feedback they would have received had they been trained on the same network configuration as the ASD participants. This served as a control for any changes in connectivity in that direction being driven by something other than the feedback. At the same time, this control also examined whether it is possible to modulate any network, regardless of the native connectivity.

**Table 1.** Exit questionnaires

| | 1. what did you think the study was about? | 2. what were you doing during the scans? (Ask about each day if applicable) | 3. how hard were you trying to solve the puzzles, on a scale of 1–10, for each day? |
|---|---|---|---|
| 1 | Really don't know. Figure out how long it takes my brain to figure out where all the pieces go. | Laying back, relaxing, thinking about random things. Same all days. | 5, 5, 5, 5 |
| 2 | MRIs, puzzle study | Day1: just think, Day 2: just think, Day 3: stay awake, Day4: stay awake | 9, 9, 9, 9 |
| 3 | Pictures and the emotional responses they elicit. Definitely tied to emotions. Thought he had to be less excited for pieces to come up | Thinking about memories, things to pass the time | 10, 10, 9, 8 |
| 4 | Measuring the thought processes of where certain processes take place in the game. Stress test | Day1: eye movement, control heart rate, close eyes; Day2: nothing, relaxed; day3: relax, day4: nothing | 5, 1, 1, 2 |
| 5 | About focus | Trying to focus on certain areas of the screen | 5, 8, 10, 2 |
| 6 | What exactly is autism and how the brain is related to it? The puzzles can show how fast different people's brains work. | Nothing different really. Today was thinking about an English assignment. Was really just looking at them, wanted to see the picture. Didn't think 'oh I have to do this now'. Wasn't trying hard to solve at all. Either it comes or it doesn't. You can't really rush your brain | 1, 1, 1, 1 |
| 7 | I don't know | Breathing in patterns, blinking, gave up | 7, 4, 5, 10 |
| 8 | About first level visual processing? | Most math, computation | 5, 2, 2, 2 |
| 9 | Multitasking, and being able to keep still and look in the same place for a consistent period. | Tried different ways to solve more pieces. Today, for example, tried to keep still and look at a similar spot consistently. Yesterday, tried multitasking, breathing and blinking on different rhythmic scales (helped the most). Last week, kept looking around in various directions, mostly at squares that werent filled in yet. | 6, 6, 7, 8 |
| 10 | How humans see stuff and how the brain reacts. Ie sight | Staring at whatever space he could to put the puzzle together | Same each day |
| 11 | Don't know, detecting how hard my brain works on puzzles | Trial and error, look at different parts of screen, try different techniques | 9, 7, 9, 8 |

DOI: https://doi.org/10.7554/eLife.28974.008

## Learning

To assess whether any learning took place over the course of these initial four training days, we examined the correlations between the two target regions (which had been trained to increase connectivity), the two target-control pairs (which were trained to decrease connectivity), as well as the composite difference measure. *Figure 3* shows the results of this analysis for the ASD group. As can be seen, over the course of the four training days, correlations between the two target regions steadily increased (with a significant difference between most days, $p=4\times10^{-4}$ between day1 and day4, mean change in correlation = 0.11, *Figure 3A*), while correlations between target1 and control decreased (significant difference between day1 and all other days, $p=8\times10^{-4}$ between day1 and day4 mean change = −0.13, *Figure 3B*). Though there is an overall decrease between target2 and control, this does not reach significance and the mean change is small (*Figure 3C*). This is in line with the lack of differentiation between the ASD group and the TD group in target2-control correlations (*Figure 1C*), suggesting neurotypical network connectivity is more resilient to change. *Figure 3D* shows the overall composite difference measure, taking into account all three correlation pairs, where there is a strong and consistent increase between day1 and day4 ($p<2\times10^{-4}$, mean change = 0.19). 15 of the 17 participants showed a positive change in this measure (14/17 had a positive change in target1-target2 correlations, as well as a negative change in target1-control correlations). The TD control group on the other hand, showed no significant changes between days in any of the three pairwise combinations, or in the composite difference measure. Only 4/10 participants in this group showed a change in the trained direction in the composite measure, within the range of chance, and the magnitude of change was minimal relative to the change seen in the ASD participants. *Figure 4A* shows these data for all the individual ASD participants, while *Figure 4B* shows the individual TD participants. The full results for all individual participants, for all days, are displayed in

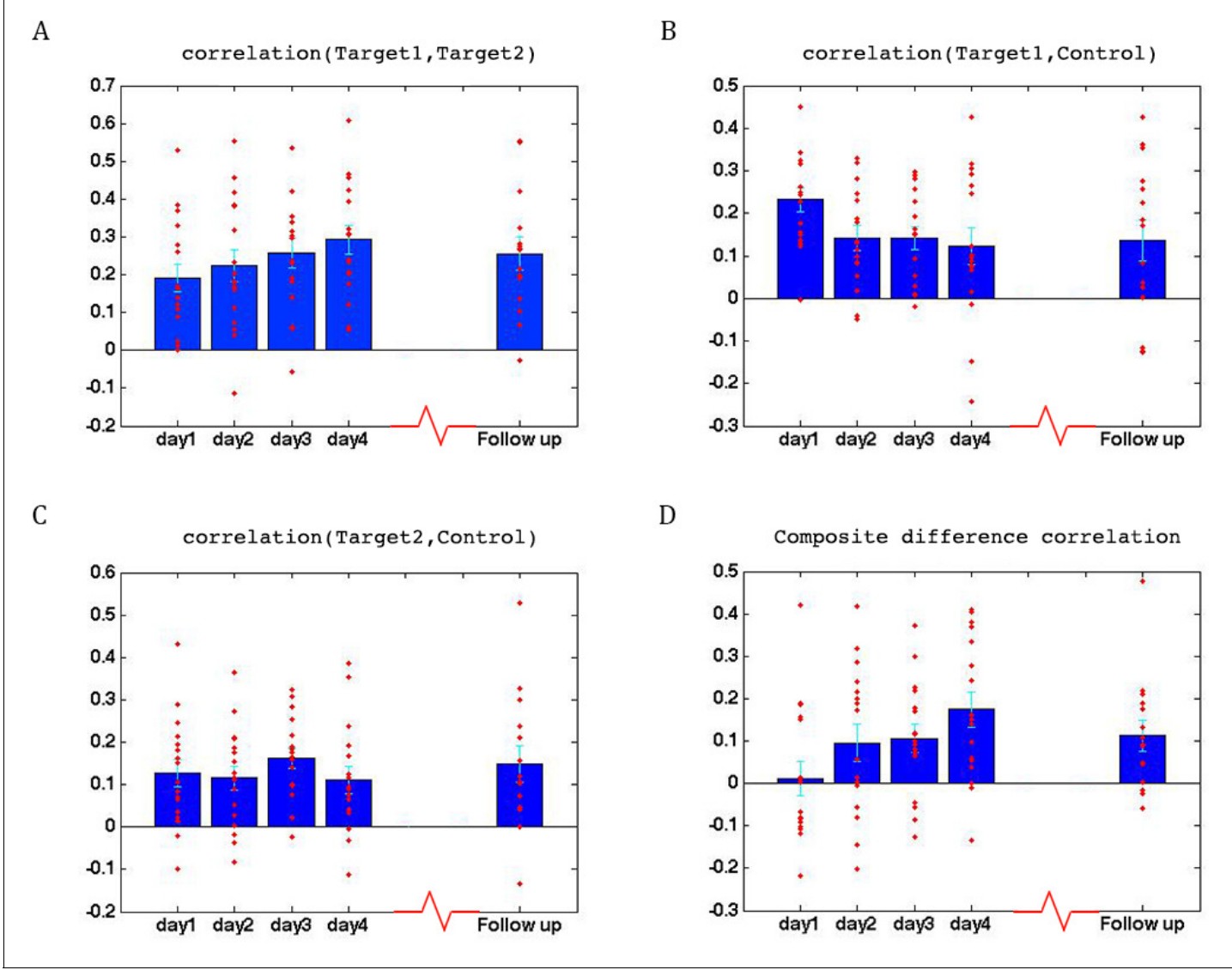

**Figure 3.** Learning across days, ASD group. (A) Correlations between the two target regions per day, averaged across all four neurofeedback scans per day. Blue bars represent the subject mean, cyan error bars mark ± SEM. Red dots represent each individual subject. The difference in correlations between day1 and days 3 and 4 and the follow up is significant (p=0.05, p=4×10$^{-4}$, p=0.05 respectively), as is the difference between day2 and day4 (p=0.016). (B) Correlations between Target1 and Control. There is a significant change between day1 and all other days (p=0.005, p=5×10$^{-4}$, p=8×10$^{-4}$ for days 2, 3 and 4, p=0.015 for the follow up). (C) Correlations between Target2 and Control. (D) Composite difference measure, showing the difference between target-target and target-control correlation pairs (see Materials and methods). Day1 correlations are significantly different from all other days (p=0.003, p=0.01, and p=2×10$^{-4}$, p=0.007 respectively), and day2 is significant different from day4 (p=0.0017). In all panels N = 17 for days 1–4, N = 15 for the follow up. All p-values for differences between days were determined by permutation tests.

DOI: https://doi.org/10.7554/eLife.28974.009

The following figure supplements are available for figure 3:

**Figure supplement 1.** Retention of learning.
DOI: https://doi.org/10.7554/eLife.28974.010

**Figure supplement 2.** Data from first two neurofeedback scans only.
DOI: https://doi.org/10.7554/eLife.28974.011

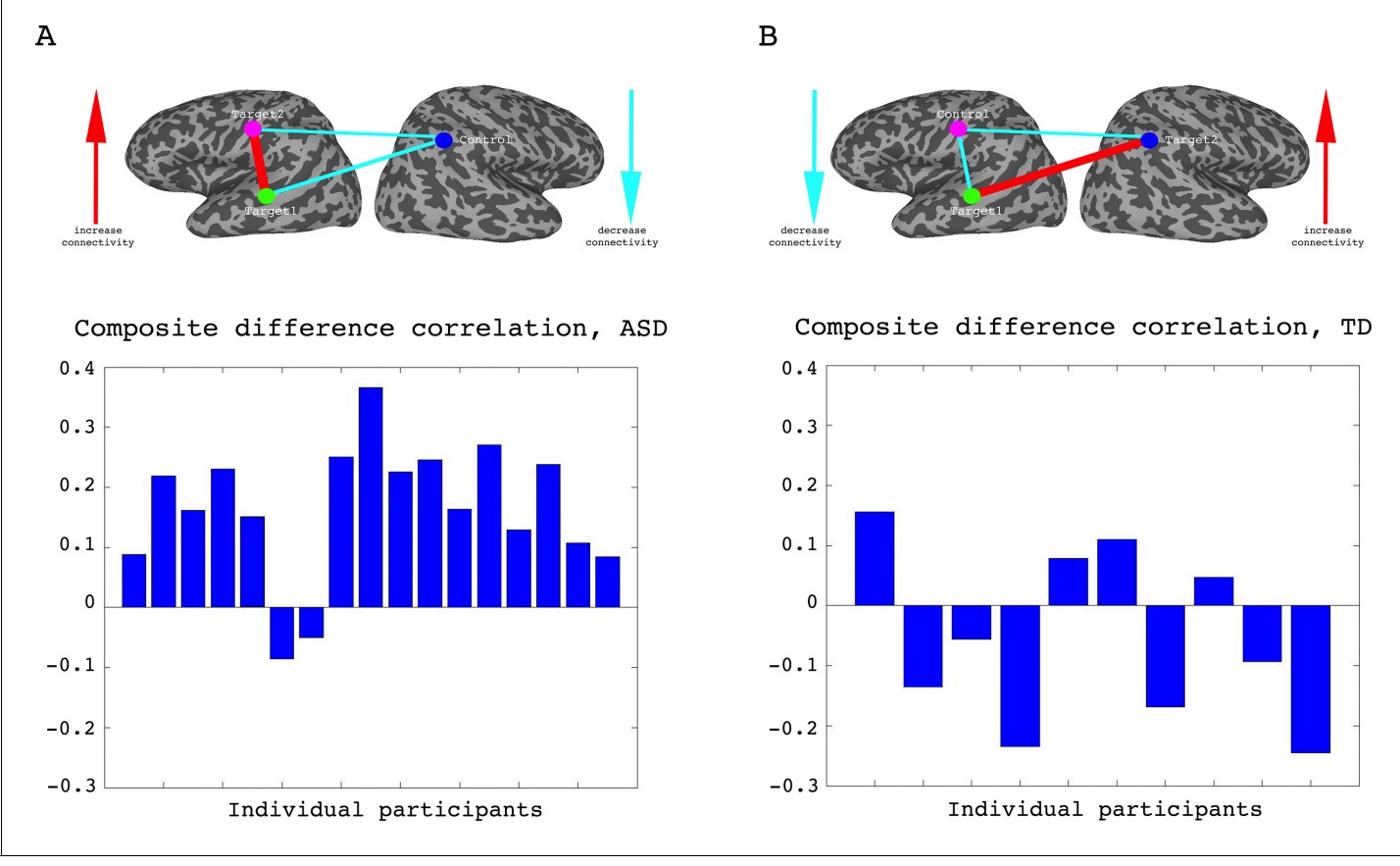

Composite difference correlation, ASD

Composite difference correlation, TD

**Figure 4.** Individual participant data. (**A**) Difference in the composite difference correlation from day1 to day4, for each of the 17 individual ASD participants averaged across all four neurofeedback scans per day, presented chronologically in order of scanning. (**B**) Same analysis for each of the 10 TD participants, presented chronologically. Note that the composite difference measure is comprised of target-target correlations minus the target-control pairs, and that the definition of targets and control differed between the two groups.

DOI: https://doi.org/10.7554/eLife.28974.012

The following figure supplement is available for figure 4:

**Figure supplement 1.** Individual participants.
DOI: https://doi.org/10.7554/eLife.28974.013

*Figure 4—figure supplement 1*. To further ensure that the null result in the TD group was truly different from the significant result in the ASD group, and not simply due to lower statistical power because of the smaller sample size, we re-calculated the change in correlations for each of the possible subsets of 10 participants from our ASD group, and found that the significant difference between day1 and day4 was maintained for all subsets in the target-target correlations (range of mean change 0.03–0.17, p-value range 0.046 to $1 \times 10^{-4}$), for 83% of possible subsets in the target1-control correlations range of mean change −0.02 to −0.19, $0.22 < p < ,1 \times 10^{-4}$ and for all possible subsets in the composite correlation difference measure (range of mean change 0.1–0.23, p<0.007 for all subsets). To test for interaction between the TD and ASD groups, we carried out a two sample t-test (without assuming equal variance to control for different sample size) between the change in composite correlations in the ASD group vs. the change in composite correlations in the TD group. A significant difference (p=$9.2\times10^{-4}$) was found between the learning seen in the TD and ASD groups in this analysis. This significant difference between the groups was maintained even when we chose the ASD subset of 10 participants with the smallest change (p=0.013).

We next set out to test how long this learning would be maintained. To address this question, we called back the participants for a follow up study, in which they returned for another, shorter round of training. To get a good indication of the persistence of the training effect, participants were called back in a staggered manner, from as little as 5 weeks and up to 56 weeks from their original training.

Our results indicate that the learning was mostly, though not fully, preserved, even after such an extended time period (*Figure 3A–D*, follow up). In fact, although there was variation between subjects in the degree of retention, there was no correlation between the time that had elapsed and the rate of retention (see *Figure 3—figure supplement 1*). Since there were only two feedback runs in this follow up scan, we also compared them to just the first two feedback runs for the first four days, in order to account for any differences arising from the different number of runs. The results using just the first two runs for the first four days were not in any way different from the results using the full data (see *Figure 3—figure supplement 2*).

## Whole brain analysis

So far we had only considered what happens in the regions that were trained. In order to get a more comprehensive picture of the effects of the training on the brain, we conducted a whole brain analysis, which looked for changes during the training period (i.e. from day1 to day4). We calculated three maps, one for target1, one for target2, and one for the control region, with each map showing the change from day1 to day4 in the correlation of each voxel in the brain to the corresponding region. We then carried out a t-test across all participants for each of these three analyses, and the resulting maps for the ASD group are displayed in *Figure 5*. The changes were exactly as predicted by the training: the strongest positive change in correlation to target1 over the training period was in the somatosensory cortex (with a peak at target2), and the strongest negative change was in the control region. Changes to correlations with target2 were seen in the STS with a peak in target1, and negative changes in correlation to control were seen in bilateral STS (*Figure 5A*). Since we were training a network of three nodes, rather than a simple connection between two regions, we next calculated the composite change: for each voxel, the change between day1 and day4 in its correlation to target1 minus its correlation to control (*Figure 5B*), and the same change in its correlation to target2 minus control (*Figure 5C*). This analysis yielded similar but far stronger results. The maps of the composite correlations were corrected at a very conservative cluster threshold determined by random permutation testing, in accordance with recent statistical recommendations for analyses utilizing cluster size (*Eklund et al., 2016*) (see Materials and methods). These results support a causative role for the feedback itself, as the specific relationship that was trained between the two targets and the control came up in completely independent, whole-brain analysis. That is, using target1 relative to the control seed, the largest change in the whole brain was found in target2, even though that region was not pre-selected and the analyses did not constrain this to happen, and vice-versa using target2 and control. Note that we do not expect to find changes between day1 and day4 in either target1 or control in the target1-control map, as these regions did not change in relation to themselves. Rather, this analysis highlights all the other areas, outside of those two regions, which changed their correlation over the course of training in relation to target1 (increasing) and to control (decreasing), finding the peak of this change in target2. The same is true for the target2-control map, which shows an even greater effect focused on target1, consistent with the ROI analysis results showing a greater decoupling of target1 from control than target2 from control. Note that *Figure 5* shows results only for the ASD group, as no significant peaks were identified in any of the target or control regions for the TD control group, and no voxels survived the cluster correction threshold.

## Transfer to resting state following training

The training-related changes we have demonstrated to this point were during the neurofeedback scans themselves. To be of any potential clinical value, these changes must also generalize beyond the training sessions, to the resting state scans, which reflect the baseline connectivity of the brain when not engaged in a specific task. In order to obtain as accurate an estimate of baseline as possible and to avoid any contamination by the task, only the two rest scans prior to the neurofeedback were used for this analysis. Changes were overall smaller than those seen during the training, but significant changes were found between day1 and day4 (target1-target2 correlations mean change = 0.07, p<0.038, composite correlations measure mean change = 0.1, $p<2\times10^{-4}$), and between day1 and the follow up (target1-target2 mean change = 0.09, p<0.011, composite correlations measure mean change = 0.11, $p<2\times10^{-4}$). Change in rest was significantly correlated to change during the neurofeedback scans (r = 0.42, p<0.04, permutation test). Moreover, 14/15 participants who came in for the follow up showed an increase in the composite correlation measure

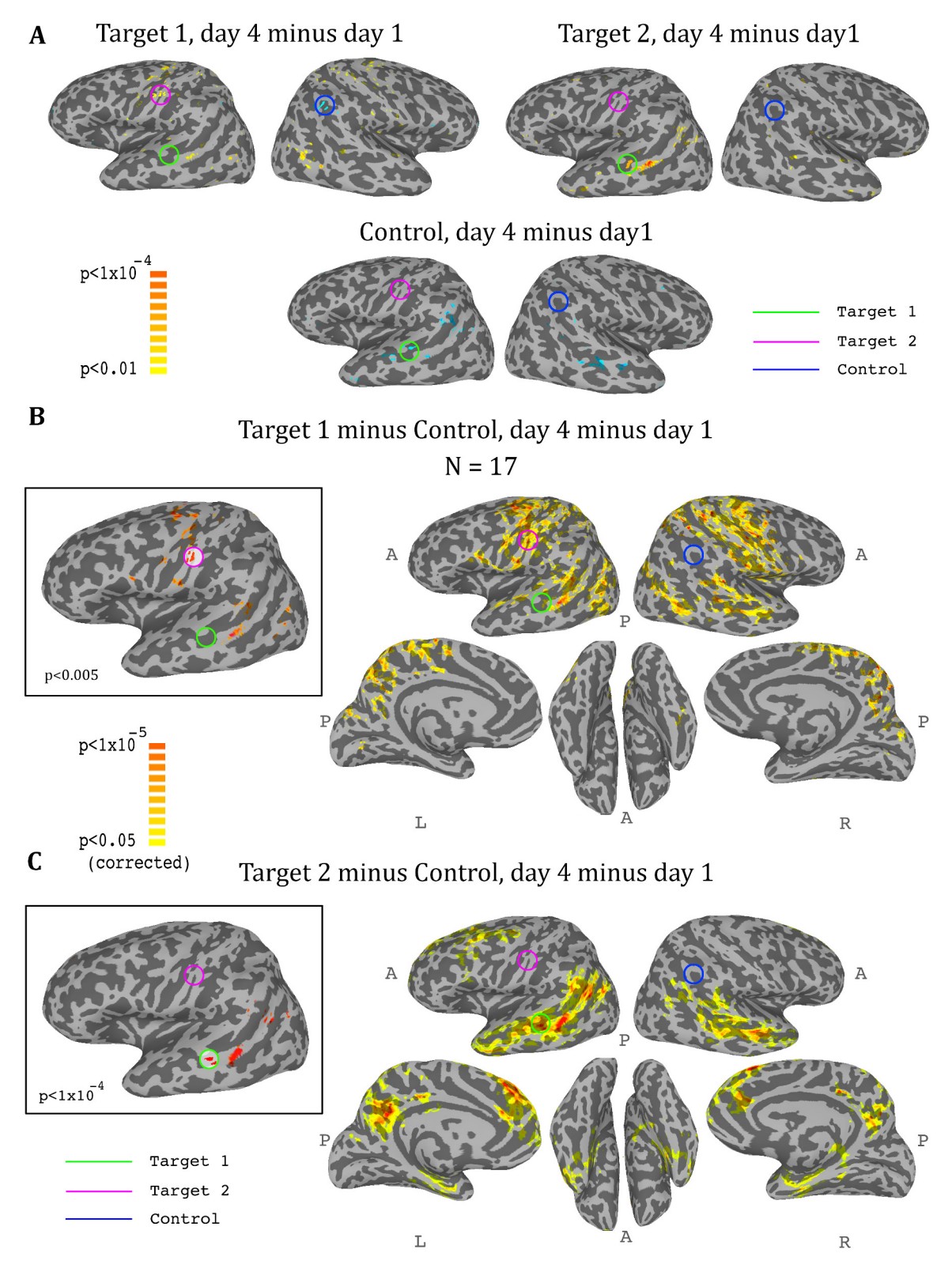

**Figure 5.** Whole-brain analysis during neurofeedback, ASD group. (**A**) Top left: change in correlations to target1, between day1 and day4, t-test across participants. High values represent voxels that showed a consistent change between day1 and day4, such that on day4 they were more correlated to target1 than they were on day1. Note the positive peak in target2, and the negative peak in the control region. Top right: change in correlations to target2, between day1 and day4, t-test across participants. Note the positive peak in target1. Bottom: change in correlations to the control region,
*Figure 5 continued on next page*

*Figure 5 continued*

between day1 and day4, t-test across participants. Note the negative peak in target1 and bilateral STS. (B) Change in differential correlation to the target1 and control ROIs, between day1 and day4, t-test across participants. High values represent voxels that showed a consistent change between day1 and day4, such that on day4 they were more correlated to target1 and less correlated to control than they were on day1. Inset shows the same analysis at a higher threshold. (C) Same as (B), for target2 and control. Note that for both maps, the other target, which was not included in the analysis, emerges as the area of greatest change across training days. Maps corrected through permutation tests (see Materials and methods).
DOI: https://doi.org/10.7554/eLife.28974.014

(*Figure 6A*). To assess whether the changes seen in the follow up could simply be a function of the elapsed time, we examined data from all participants in the previous study (used to define the training regions) which had at least two resting state scans from two different time points, and evaluated the change in connectivity seen between the sessions. 19 participants had two such data points, and the average time between the sessions was 13.2 months. The change however was not significant for any of the pairwise correlations, or the composite correlation measure (*Figure 6A*). We next looked at the composite measure for the 10 participants from our study who had also participated in the previous resting state experiment (and in the follow up), and compared change from the previous experiment to the first rest sessions on day1, before any training, and in this subset also the change from day1 of the training to day4 and to the follow up (*Figure 6B*). While there was no significant change from the previous experiment to day1 (mean time interval = 38.3 months, mean change = −0.04), there was significant change to day4 (mean change = 0.1, p<0.019) and to the follow up (mean change = 0.11, p<0.003). Taken together, these analyses provide strong evidence that

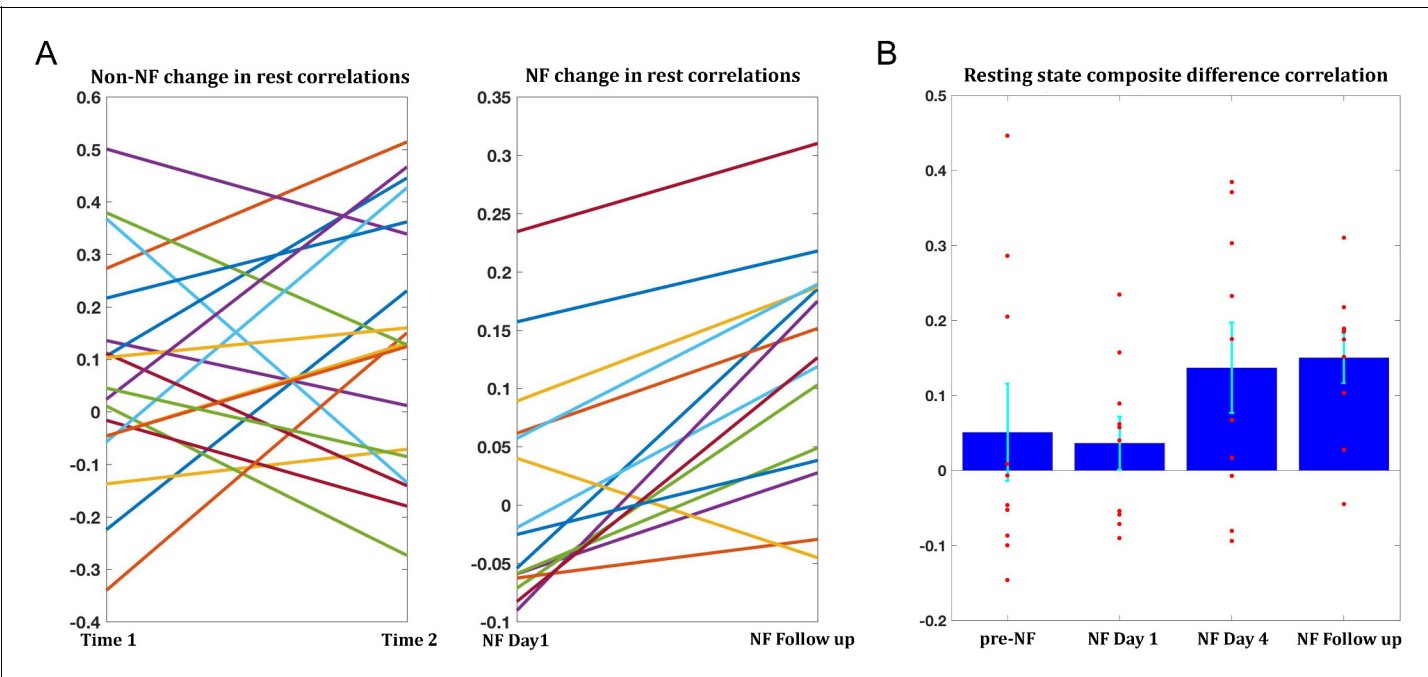

**Figure 6.** Changes in resting state correlations. (A) Left panel shows the changes in resting state composite difference correlations for the 19 participants for which two previous data points were available from a previous study, prior to neurofeedback (average time between sessions 13.2 months). The right panel shows the change in resting state composite difference correlations from the very first pre-training rest sessions on the first day of neurofeedback, to the rest sessions collected in the follow up session (also before the neurofeedback training sessions that day), for the 15 participants who took part in the follow up session. Average time between sessions for this group was 6.2 months. (B) Changes in resting state composite difference correlations for the 10 participants who had data from both the previous study, and the follow up session. Change between neurofeedback day1 and neurofeedback day4 as well as neurofeedback follow up, are significant (p=0.019, p=0.003 respectively). All correlations are taken from the resting state scans at the beginning of the relevant session. NF = neurofeedback.
DOI: https://doi.org/10.7554/eLife.28974.015

the changes we observed were not a function of elapsed time but rather occurred as a direct result of our neurofeedback regime.

## Behavioral relevance

Finally, we asked whether the changes we see as a result of the training are in any way correlated to behavior. To this end, we looked at changes in behavior as measured by the behavioral questionnaires filled out by the parents prior to training, and two weeks after the end of the initial training set. These behavioral results included two statistical outliers who were removed from the analysis (definition of outliers was based on change scores greater than 3 standard deviation from that seen in an independent data set, see Materials and methods). We compared the change in these behavioral questionnaires to the change in correlations between the final resting state scans on the last day (following neurofeedback), and the resting state scans on the first day. There were two measures of behavior: the first, the Social Responsiveness Scale (SRS), has previously been found to correlate with functional connectivity in this network (see section on target selection), and therefore the change in this rating was expected to correlate with the change in the network. The second, the Behavior Rating Inventory of Executive Function (BRIEF), measures executive function rather than social abilities, and though patients with ASD show deficits on this measure (*Baron et al., 2000*), it is expected to reflect prefrontal functioning, and we expected that changes on this measure would not correlate to changes in the social network being trained (*Anderson et al., 2002*; *Anderson et al., 2005*; *Mahone et al., 2009*). Indeed, although there was no significant change in the mean SRS score across participants, with the mean score actually slightly (though not significnatly) increasing instead of decreasing as predicted (mean SRS before training = 69.8, mean SRS after training = 71.6, p=0.11), there was a significant correlation between changes in the resting state network and the change in SRS (pre training minus post training, so that a positive change corresponds to a reduction in symptoms, r = 0.56, p=0.016, *Figure 7*). There was no significant correlation between the baseline SRS score and the change in SRS seen during training (r = 0.14, p=0.29), or the baseline rest and the change in rest (r = −0.22, p=0.2). The partial correlation between change in rest vs. change in SRS, after controlling for baseline SRS and baseline rest, was higher (r = 0.62 p=0.014).

No such correlation was found with the BRIEF (change in SRS vs. change in resting state correlations: r = 0.56, p=0.016; change in BRIEF vs. change in resting state correlations: r = 0.09, p=0.39). We further tested for the correlation between the change in SRS vs. change in the resting state correlations, after partialing out the contribution of the BRIEF. After removing the variance explained by the BRIEF, the correlation between the rest and SRS was even higher (r = 0.6, p=0.02), indicating that change in resting state correlations was captured by the change in SRS scores, but not the BRIEF. Note that the SRS was chosen as our behavioral measure because it has consistently been shown to correlate with aberrations in network structure in ASD (*Gotts et al., 2012*; *Wallace et al., 2012*; *Di Martino et al., 2009b*; *Anderson et al., 2011*). Nevertheless, it was not designed to be used with such a short test-retest interval (3 weeks), and has only been validated for intervals of 6 months or more (*Hus et al., 2013*; *Constantino et al., 2003*; *Bölte et al., 2008*). We can therefore make no claims regarding the absolute values of the change in scores.

In the interest of completeness, we also ran the analysis with the outliers (marked in red in *Figure 7*). Without removing outliers, the correlation with SRS fell below significance, though the trend was still in the same direction (r = 0.26, r = 0.31 after controlling for baseline SRS and baseline rest, p=0.09). A possible confound in using test-retest reliability on other data to calculate outliers (as was done here) is that our neurofeedback data could include large, training induced changes, which would not be present in other datasets, skewing the variance. Note however that the two outliers reported here had a large negative behavioral change, which is not the predicted direction of change due to neurofeedback based training. This is more likely due to the short assessment interval used here (two weeks), which would give a lot of weight to negative incidents which occurred during this time. For instance, one of the participants whose data was determined to be an outlier, had moved to a new house during this time period.

## Discussion

The study of rt-fMRI covert neurofeedback – feedback given on pre-specified brain activity, but without providing participants with further information regarding the nature of the task or an explicit

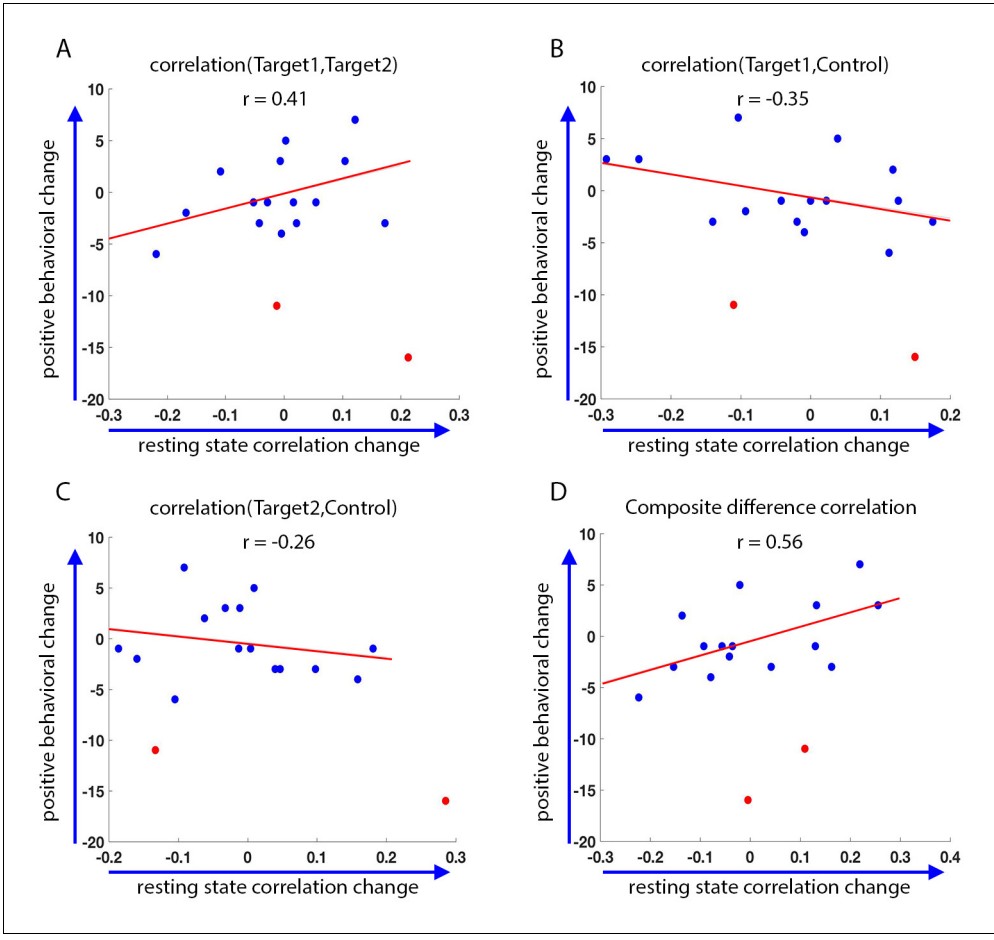

**Figure 7.** Behavioral changes. Correlation between the change in the behavioral measure (SRS) score before and after training, and the change in resting state connectivity from the post-training rest scans on day4 to the rest scans on day1 (day4 correlations minus day1 correlations, positive change corresponds to higher correlations on day4). Behavioral change on the y-axis calculated as pre-training overall SRS t-score minus post-training SRS t-score. Positive behavioral change corresponds to a reduction in SRS score after training. Statistical outliers marked in red. (**A**) Behavioral change vs. change in Target1-Target2 correlations. (**B**) Behavior vs. Target1-Control. (**C**) Behavior vs. Target2-Control. (**D**) Behavior vs. composite difference corr. R values represent Pearson's correlation. Regression lines and R values do not include the outliers. Correlations between Target1-Target2 and Target1-Control changes and behavior are in the expected direction but are not significant (p=0.07, p=0.08). Behavior vs. composite difference change is significant (p=0.016).

DOI: https://doi.org/10.7554/eLife.28974.016

strategy through which to control the feedback – is still in its infancy (for a review see [*Sitaram et al., 2017*]). These results not only help solidify the existing evidence that reward-mediated learning through covert neurofeedback is possible (*Shibata et al., 2011*; *Ramot et al., 2016*; *Amano et al., 2016*), but also expand our knowledge in important ways. In this study, we have demonstrated that covert neurofeedback can be used to modulate correlations between distinct, physically distant networks (*Figure 3*), in the great majority of participants (15 of17, *Figure 4*, *Figure 4— figure supplement 1*). We have further shown that this modulation is possible even in cases of aberrant network structure, in clinical populations, and is sustainable for extended periods of time (some of our follow up sessions were a year after the original training, and we saw no evidence for an effect of time as a modulator of retention, *Figure 3—figure supplement 1*). This kind of connectivity-based neurofeedback has previously been carried out in only a handful of studies (*Scharnowski et al., 2014*; *Megumi et al., 2015*; *Koush et al., 2017*), though the network approach

to neurofeedback is gaining traction and shows not only good preliminary findings but also has a sound theoretical basis (*Bassett and Khambhati, 2017*).

It could be argued that the changes in functional connectivity that we found were not a result of the feedback, but rather of the multiple fMRI visits, or were somehow precipitated by the nature of the puzzle task. Though imperfect because of the different population, the control using the TD group, which received feedback largely orthogonal to the network trained in the ASD group, provided further evidence for the necessity of the feedback itself in inducing these changes. The TD group, which went through the exact same protocol as the ASD group but received feedback on a different network, did not demonstrate the changes in connectivity seen in the ASD group (*Figure 4*, *Figure 4—figure supplement 1*). Moreover, the extraordinary specificity of the changes revealed by the whole brain analysis (*Figure 5*), peaking exactly at the small ROIs that were chosen for the training, would also seem to preclude an alternative explanation. The further significance of the whole-brain analysis is that this learning is spatially specific even when training disparate networks, meaning that it is possible to target specific regions of the brain. However, it is important to note in this regard that though the peaks were centered on the regions we were training, the changes spread to entire networks, as would be expected from the architecture of the brain, which is composed of large-scale networks of multiple brain regions, making it difficult to induce changes to just one region in isolation.

In a larger context, the failure to induce change in the direction of training in the control subjects, suggests that while it is possible to train networks that are fundamentally connected, it is much more difficult to train networks that are uncorrelated or weakly correlated in the typically developing brain. This conclusion is also bolstered by the failure of the training to induce change in the ASD group between target2 and control, regions which did not differ in their connectivity from that of the TDs. It is also possible that some networks may be more difficult to train than others, as has previously been suggested (*Harmelech et al., 2015*). Future studies will be needed to better understand the basic constraints in modifying these relationships.

The prohibitive cost of rt-fMRI, or fMRI scans in general, limits the number of training sessions in these paradigms. Moreover, the under-connectivity between the target regions chosen here explains only some of the behavioral deficits, and is clearly not the sole underlying cause of autism. This study was therefore designed as a proof of principle that aberrant connectivity can be addressed through neurofeedback, rather than as a clinical intervention, and whatever behavioral effects we found were expected to be modest. Once such a causal relationship is established, future potential clinical applications might pursue more cost effective options, such as identifying EEG signatures that correspond to activity from these areas, as several groups have already begun to develop (*Zotev et al., 2014*; *Meir-Hasson et al., 2014*).

From a clinical perspective, the most important result in this study is the successful transfer of the change in correlations to the resting state. By itself, change during training does not guarantee generalization of the learning, or in this case of the change in the network structure. Though the change in the baseline resting state (before neurofeedback training each day) was somewhat more modest than the change seen during training, it was reliable and consistent, and could not be explained simply by the passage of time (*Figure 6*). Note that the correlation seen with behavior (*Figure 7*) is with the change from rest on day1, to the resting state data collected immediately after the last training session on day4. This resting state change was overall smaller than the change seen with the pre-training baseline rest on day4 shown in *Figure 6*, but the day4 pre-training baseline did not show a correlation with behavior. The behavioral results, though they did not show an overall mean change, are preliminary in their scope and limited by the timescale on which they were measured, demonstrate that change in behaviorally relevant networks correlates with change in behavior, which is a crucial and entirely non-trivial point in terms of the potential clinical applications of neurofeedback. The lack of overall mean change together with the correlation with connectivity changes makes it unlikely that this result is driven by a placebo effect, and on debriefing after all data were collected, it was clear that the patients and their families did not conceive of this as an intervention, and had no expectation of behavioral change following the study. This finding also adds to the debate regarding the nature of the functional and structural hypoconnectivity found in ASD, whether it plays a causative role in ASD or is simply a downstream effect (*Vasa et al., 2016*). A change in behavior following a change in connectivity suggests the former, bolstering the mechanistic approach to functional connectivity in ASD. However, it should be noted that underconnectivity is not the full story in

ASD, and there is growing evidence for hyper cortico-thalamic connectivity, alongside the cortico-cortico hypoconnectivity (*Picci et al., 2016*). It should also be noted that the under-connectivity seen in cortico-cortico correlations in ASD is dampened connectivity (closer to 0), and not anti-connectivity (negative correlation). The nature of the feedback given here is that positive correlations between the two targets are rewarded, thus boosting connectivity that already exists at baseline (as baseline connectivity is positive, just small). Future studies will have to replicate and expand on the behavioral findings, but this method of testing behavioral changes following connectivity changes could be a promising tool for assessing different models of autism as well as other neuropsychiatric disorders.

The targets used in this paradigm were derived from a group analysis, and were not individually localized. As the variance within groups suggests, this is likely not the optimal method for ROI selection, especially if the focus is on behavioral change. These results are therefore probably an under estimation of what this tool can do, with individually tailored ROIs. It is not clear what the best method for individual ROI selection would be, or which localizer would best identify target regions, but it is another direction that should be pursued in future studies.

Since the lack of an explicit strategy allows covert neurofeedback to be used to directly target all manner of abstract, behaviorally relevant networks, potential applications could be far ranging, encompassing many clinical disorders with underlying aberrant connectivity at their core. Moreover, this is a promising technique to be used in more basic science questions, as a tool to investigate questions of causality.

## Materials and methods

### Participants

19 Males aged 15–25 (mean age = 20.93) who met the DSM-IV criteria for autistic disorder, an autism cut-off score for social symptoms on the Autism Diagnostic Review (ADR) and/or an ASD cut-off score from social +communication symptoms on the Autism Diagnostic Observation Schedule (ADOS), all administered by a trained, research-reliable clinician, were recruited for this experiment. Additionally, 11 age matched typically developing males were recruited for the control group. All participants had normal to corrected to normal vision. IQ scores were obtained for all participants, and all full-scale IQ scores were ≥85 as measured by the Wechsler Abbreviated Scale of Intelligence, the Wechsler Adult Intelligence Scale-III, or the Wechsler Intelligence Scale for Children-IV. Participant groups did not differ in terms of full-scale IQ.

one ASD participant was removed due to discomfort in scanner on day1, and another ASD participant was removed on day two due to anxiety. 1 TD participant was removed after day1 for excessive motion. 17 ASD and 10 TD participants completed all four days of neurofeedback training. 15 ASD participants returned for the follow up experiment. The experiment was approved by the NIMH Institutional Review Board (protocol 10_M-0027). Written informed consent was obtained from all participants.

### Definition of ROIs

Three Regions of Interest (ROIs) were selected for training: two targets and one control. The targets were chosen according to previous research as those with a large degree of reduced connectivity in Autism Spectrum Disorder compared with typically developing (TD) controls, based on between group analysis as explained in (*Gotts et al., 2012*). For this analysis, we used an expansion of the dataset published in (*Gotts et al., 2012*), N = 56 ASD, 62 TD. Of the 56 ASDs in this dataset, 11 participated in the neurofeedback study. Additional constraints placed on the choice of ROIs was for them to be physically distant from each other, and in different networks (see (*Gotts et al., 2012*) for details). All ROIs were defined as spheres of 4 mm radius surrounding the focal points: Target1 - left Superior Temporal Sulcus (Talairach coordinates: −49,−29, 0), Target2 - left somatosensory cortex (Talairach coordinates: −54, 14, 39), and Control - right Inferior Parietal Lobe (chosen to be as uncorrelated with these two targets as possible in the TD dataset, Talairach coordinates: 49,−50, 42). *Figure 1* shows the between group difference in the correlations between the ROIs.

## Imaging data collection and MRI parameters

All scans were collected at the Functional Magnetic Resonance Imaging Core Facility on an 8 channel coil GE 3T (GE Signa HDxT 3.0T) magnet and receive-only head coil, with online slice time correction and motion correction. The scans included a 5 min structural scan (MPRAGE) for anatomical co-registration, which had the following parameters: TE = 2.7, Flip Angle = 12, Bandwidth = 244.141, FOV = 30 (256 × 256), Slice Thickness = 1.2, axial slices. EPI was conducted with the following parameters: TR = 2 s, Voxel size 3.2*3.2*3.2, Flip Angle: 60, TE = 30 ms, Matrix = 72×72, Total TRs = 270, Slices: 37. All scans used an accelerated acquisition (GE's ASSET) with a factor of 2 in order to prevent gradient overheating.

## Neurofeedback experiment

The initial neurofeedback experiment consisted of 4 training sessions over 8 days. There were 2 consecutive training days, a 6 day delay, then a final set of 2 consecutive training days. Each training day had 2 initial rest scans, 4 neurofeedback sessions, and 2 final rest scans. All scans were 9 min long. Participants were instructed to maintain an eyes-open rest and look at the blank screen. Neuropsychological tests were administered at two timepoints: on the first training day before scanning, and two weeks following the last training day.

## Follow-Up experiment

Follow up scans were conducted 5–56 weeks after the final training day and consisted of a single, abbreviated neurofeedback session with two rest scans followed by two neurofeedback sessions.

## Online real-time data collection

Regions of Interest (ROIs) were defined in Talairach space as described above. The standard Talairach brain was then co-registered to the structural scan collected that day, which was in turn co-registered to a short (10 TRs) functional echo-planar imaging scan (setup EPI) collected for that purpose each day before the first resting state session, to bring the ROIs into the native space during neurofeedback processing. All coregistration was carried out with the AFNI (Analysis of Functional Neuro-Images) software package (*Cox, 1996*).

## Real-time fMRI algorithm

During online processing of the data, 3D motion correction and slice time correction were carried out on all functional images. BOLD signal was extracted from each voxel in the ROIs and the mean signal was calculated for each ROI.

Feedback decisions were determined by a difference measure, taking into account both the changes in the trend between the two target ROIs and the control ROI. This difference measure was calculated for each TR and for each of the three ROIs. Our rt-fMRI algorithm calculated the difference between the mean signal in the current TR minus the signal in the preceding TR, giving the signal trend in each ROI (increasing or decreasing). If the trend in the two targets was the same, and opposite from the trend in the control ROI, then feedback was given, meaning both conditions had to be fulfilled for feedback to be given:

$$\frac{ms(Target1(TR=t)) - ms(Target1(TR=t-1))}{ms(Target2(TR=t)) - ms(Target2(TR=t-1))} > 0 \ \& \ \frac{ms(Target1(TR=t)) - ms(Target1(TR=t-1))}{ms(Control(TR=t)) - ms(Control(TR=t-1))} < 0$$

(ms = mean signal)

## Neurofeedback procedure

Each training session had four neurofeedback training scans. The scans started out with a uniformly grey screen. Participants were told that there is a picture hidden underneath, and were instructed to try to unveil the image during what was described as the puzzle task. Importantly, no further cognitive strategies or suggestions were given to the participant for the duration of the experiment. Participants were not informed that their performance on the puzzle task was determined by brain activation.

## Neurofeedback stimuli

Participants received two forms of positive reinforcement whenever the real-time algorithm determined that the network requirements had been met: a 'puzzle piece', i.e. a square of the hidden picture, would become visible on the screen with a concomitant sound of positive valence. This feedback was chosen to maximize engagement with the paradigm during the scan by providing a complex and interesting visual stimulus in a game-like setting, and the auditory stimulus was paired to ensure that participants would be aware of positive feedback independent of their visual attendance.

## Visual stimuli

During rest scans participants were shown a uniformly grey screen.

During Neurofeedback training participants would begin with a uniformly grey screen. Then the image would become visible in small rectangular blocks, described to the participants as 'puzzle pieces.' There were 25 'puzzle pieces' per board, which would be displayed piece by piece until a whole image was unveiled. After a full board was completed, the screen would become blank and a new image would begin to appear. The images were randomly selected from a set of 100 non-social images devoid of people or text, like a landscape or an abstract painting.

At the end of each 9 min training round, participants viewed a scoreboard which told them how many individual pieces they had unveiled that round, as well as the top score that they had received that day. Participants were financially incentivized to beat their best score for that day. fMRI offline data preprocessing:

Post-hoc signal preprocessing was conducted in AFNI. The first four EPI volumes from each run were removed to ensure remaining volumes were at magnetization steady state, and remaining large transients were removed through a squashing function (AFNI's 3dDespike). Volumes were slice-time corrected and motion parameters were estimated with rigid body transformations. Volumes were coregistered to the anatomical scan. Volumes were smoothed with 6 mm blurring and normalized by the mean signal intensity of each voxel. The AFNI ANATICOR procedure was then applied to remove nuisance physiological and nonphysiological artifacts from the data (*Jo et al., 2010*). The anatomical scan was segmented into tissue compartments with Freesurfer (*Fischl et al., 2002*), Ventricle and white-matter masks were created and applied to the volume-registered EPI. Prior to smoothing, these masks gave pure nuisance times series for the ventricles and local estimates of the BOLD signal in white matter, averaged within a 15 mm radius. The measured respiration and heart rate signals were used to create Retroicor (*Glover et al., 2000*) and respiration volume per time (RVT) regressors (*Birn et al., 2008*). All nuisance time series in every run (average ventricle time series, average local white matter time series, 6 parameter estimates for head motion, and thirteen RVT and Retroicor regressors) were detrended with fourth-order polynomials before least-squares model fit to each voxel time series. No other filtering of the data was done. All participant data was aligned by affine registration to AFNI's TT-N27 template in standardized Talairach and Tournoux (*Talairach and Tournoux, 1988*) space.

## Neuropsychological tests

Baseline neuropsychological tests were conducted before the initial training session, and post-experiment surveys were collected two weeks after the final neurofeedback session. Parents filled out the Social Behavior Scale (SRS) to identify common social behaviors in autism, as well as the Behavioral Rating Inventory of Executive Function (BRIEF). The 'informant' report (filled in by a parent) was used as it has been shown to be more accurate (*McMahon and Solomon, 2015*). An independent dataset of ASD subjects who did not participate in this experiment but had SRS test-retest data was used to determine change reliability. Data points that were beyond three standard deviations from the mean as determined by this analysis were excluded as outliers.

## Cognitive strategy questionnaire

We developed a cognitive strategy questionnaire that was completed by 11 of the 17 participants. Following their final scan session on day4 of the training, each of these participants was asked what they thought the experiment was about. Participants were then asked what they were doing during the scans, and if they used a particular cognitive strategy.

Finally, participants were asked to rate on a scale of 1–10 how hard they had been trying to solve the puzzles each day, how satisfied they felt when a new puzzle piece came up, and if there were differences between days. The first six participants did not complete this questionnaire, but were interviewed after the final scan and reported no knowledge of the objective of the task, and similar cognitive strategies to those later reported in the questionnaire. See *Table 1* for the data from these questionnaires.

## Data analysis

All data were analyzed with in-house software written in Matlab, as well as the AFNI software package. Data on the cortical surface were visualized with SUMA (SUrface MApping) (*Saad et al., 2004*). The composite difference measure was computed by subtracting the average correlation of the two target/control pairs, from the target/target correlation:

$$corr(Target1, Target2) - \frac{1}{2}(corr(Target1, Control) + corr(Target2, Control))$$

All p-values for the changes in correlation between days were computed through permutation tests, randomly permuting the days for 5000 iterations.

## Whole-brain analysis

For each participant, for each neurofeedback scan on day1 and day4, we first transformed the correlation values with Fisher's z-transform to improve normality, then calculated a difference measure per voxel: corr(voxel time series, avg. Target1 time series) - corr(voxel time series, avg. Control time series). The resulting maps held information regarding each voxel's differential correlation to the Target1 vs. Control ROIs. We then averaged the maps for each participant across all four neurofeedback scans for each of the two days, and subtracted the average day1 map from the average day4 map. Each voxel in the resulting map now signified the change in correlation from day1 to day4, in the differential correlation to the Target1 ROI vs. the Control ROI, where a positive value means that this voxel was differentially more correlated to Target1 than to Control on day4 compared with day1. Normality of these data were ascertained using Lilliefor's goodness of fit test. We then carried out a t-test across the 17 participants, to identify voxels with a consistent change across subjects. Maps were corrected using a permutation test to determine significant cluster size, with day1 and day4 randomly permuted for each participant across 5000 permutations (as suggested by [*Eklund et al., 2016*]). These permutations were carried out at p-value thresholds of 0.05, 0.01, 0.005, 0.001 and 0.0005, and a mask was created of voxels that survived any of these corrections. The mask was then applied to the map shown in *Figure 5A*, which was set at a p-value threshold of 0.05.

The same procedure was carried out for the Target2 minus Control differential correlation, and the resulting map is shown in *Figure 5B*.

## Data availability

Data is available via the XNAT platform https://central.xnat.org/app/template/Index.vm (dataset title 'Direct modulation of aberrant brain network connectivity through real-time neurofeedback' with ID number ASD_NF). As some of the participants in the experiment signed an older version of the consent form, which does not explicitly allow for data sharing, we are currently working on re-consenting all the participants with a new version. Hence for now, users will need to request access through the system. This can be done by creating a XNAT user account and pressing the request access link.

## Acknowledgements

We thank Lauren Kenworthy for insights into behavioral testing methods in ASD, and Miriam Menken for help with data pre-processing. This work was supported by the Revson Foundation Women in Science award through the Weizmann Institute of Science (to MR) and by the Intramural Research Program, National Institute of Mental Health (ZIAMH002920 and ZIAMH002783).

## Additional information

### Funding

| Funder | Grant reference number | Author |
|---|---|---|
| National Institute of Mental Health | ZIAMH002920 | Michal Ramot<br>Sara Kimmich<br>Haroon Popal<br>Emily White<br>Stephen J Gotts<br>Alex Martin |
| National Institute of Mental Health | ZIAMH002783 | Javier Gonzalez-Castillo<br>Vinai Roopchansingh |

The funders had no role in study design, data collection and interpretation, or the decision to submit the work for publication.

### Author contributions

Michal Ramot, Conceptualization, Data curation, Software, Formal analysis, Investigation, Visualization, Methodology, Writing—original draft, Writing—review and editing; Sara Kimmich, Data curation, Investigation, Project administration, Writing—review and editing; Javier Gonzalez-Castillo, Vinai Roopchansingh, Methodology, Writing—review and editing; Haroon Popal, Emily White, Data curation, Investigation, Project administration; Stephen J Gotts, Conceptualization, Software, Validation, Writing—review and editing; Alex Martin, Conceptualization, Supervision, Funding acquisition, Writing—original draft, Writing—review and editing

### Author ORCIDs

Michal Ramot (iD) http://orcid.org/0000-0001-9716-6469

### Ethics

Human subjects: The experiment was approved by the NIMH Institutional Review Board, protocol number 10-M-0027, clinical trials number NCT01031407. Written informed consent and consent to publish were obtained from all participants. All procedures performed were in accordance with ethical standards set out by the Federal Policy for the Protection of Human Subjects (or 'Common Rule', U.S. Department of Health and Human Services Title 45 DFR 46).

### Decision letter and Author response

Decision letter https://doi.org/10.7554/eLife.28974.020
Author response https://doi.org/10.7554/eLife.28974.021

## Additional files

### Supplementary files

• Transparent reporting form
DOI: https://doi.org/10.7554/eLife.28974.017

### Major datasets

The following dataset was generated:

| Author(s) | Year | Dataset title | Dataset URL | Database, license, and accessibility information |
|---|---|---|---|---|
| Ramot M, Kimmich S, Gonzalez-Castillo J, Roopchansingh V, Popal H, White E, Gotts SJ, Martin A | 2017 | Direct modulation of aberrant brain network connectivity through real-time neurofeedback | https://central.xnat.org/app/template/Index.vm | Access can be requested via creating a XNAT user account |

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
