## [Decision Letter]

Thank you for submitting your article "Direct Modulation Of Aberrant Brain Network Connectivity Through Real-Time NeuroFeedback" for consideration by *eLife*. Your article has been reviewed by three peer reviewers, and the evaluation has been overseen by Nick Turk-Browne as the Reviewing Editor and coordinated by Rich Ivry as the Senior Editor. The reviewers have opted to remain anonymous.

The reviewers have seen and discussed all of the reviews, and the Reviewing Editor has drafted this decision to help you prepare a revised submission.

Summary:

This paper reports a real-time fMRI study seeking to alter functional connectivity in ASD. This includes both up-regulating an edge between two target ROIs that is deficient in ASD and down-regulating their respective edges to a control region, to show that the effects are not global. Moreover, the effects of neurofeedback are compared to a typically developing control group. The key findings are that: (1) connectivity between the target ROIs increases across training sessions, connectivity between the first target and the control region decreases, and a composite score tracking all edges also increases; (2) these effects carry over to resting state outside of the context of training; (3) no effects are observed in the control group; and (4) the changes in connectivity correlate with an ASD scale tracking symptoms.

Everybody is in agreement that this is an innovative and ambitious study, with very promising results and significant implications. We all also realize that the complexity of this work means that there are some potential holes that would need to be patched in future work (e.g., mismatched sample sizes, different training regimens across groups, other possible controls, etc.). That said, we are eager to move forward with this manuscript should you be able to address the reviewers' key comments below.

Essential revisions:

1) The manuscript text and figure legends do not adequately report all statistical tests, even in cases where comparisons are described as "significant" (e.g., Figure 1). The results of all tests should be reported fully, including comparisons of each group to zero and comparisons between groups.

2) Relatedly, there are many instances where the groups are compared explicitly or implicitly. Any such comparisons need to be supported by interactions/tests between groups. Given the different sample sizes, the authors may need to adopt the subsampling technique used elsewhere (or other non-parametric method).

3) Figure 7 needs to include x and y values. The authors state that the absolute value of the behavioral change is not meaningful, but then it is unclear why they excluded outliers and why they report correlations of those values (resting state connectivity should be in a standard unit anyway). Regardless of issues with interpretability (which can be noted in the legend), the reviewers felt strongly that plotting these values was a basic matter of convention related to providing a scale and baseline (e.g., 0).

4) The correlation between SRS and target ROI connectivity at pre-training is potentially problematic, as these behavioral and connectivity scores provide the baseline for the difference metrics used to assess the relationship of brain-behavior changes (Figure 7). How does baseline behavior/connectivity contribute to this change relationship? What if you partial out or control for the baseline?

5) The SRS could be biased by demand characteristics on the part of the parents. How can the authors rule out the possibility of contamination from, for instance, the parents' expectations of positive behavioral change due to neurofeedback training? This limitation should be acknowledged and more details should be reported on the mean SRS before and after training, and the reliability of the difference.

6) The exclusion of outliers in the SRS change score based on test-retest reliability is problematic. A large change could reflect neurofeedback-based learning that did not (and could not) contribute to basic test-retest reliability in other data. The relationships should be reported with and without the outliers, with a more circumspect discussion about the justification for the exclusions.

---

## [Author Response]

Essential revisions:1) The manuscript text and figure legends do not adequately report all statistical tests, even in cases where comparisons are described as "significant" (e.g., Figure 1). The results of all tests should be reported fully, including comparisons of each group to zero and comparisons between groups.

We thank the reviewers for pointing out this omission. We have amended the text and legends to include the results of all statistical tests (subsections “Selection of training targets” and “Behavioural relevance”, figure legends 1, 3, 6 and 7).

2) Relatedly, there are many instances where the groups are compared explicitly or implicitly. Any such comparisons need to be supported by interactions/tests between groups. Given the different sample sizes, the authors may need to adopt the subsampling technique used elsewhere (or other non-parametric method).

This is an important point. The comparison between groups for the ROI selection analysis was conducted through a permutation test preserving original group size, to account for the difference in number of participants. This is now better explained in the text (subsection “Selection of training targets”). We also carried out an additional analysis, directly comparing group interaction for learning in the TD and ASD group, through both a two sample t-test without assuming equal variance, on the full dataset, as well as a sample-size matched two sample t-test between just the 10 worst learners in the ASD group, and the TD group. A significant between group difference was found in both cases (subsection “Learning”). These tests compared the interactions between the groups, by directly comparing the learning (i.e. changes in correlations) between the groups, which is equivalent to the interaction in a repeated measure anova.

3) Figure 7 needs to include x and y values. The authors state that the absolute value of the behavioral change is not meaningful, but then it is unclear why they excluded outliers and why they report correlations of those values (resting state connectivity should be in a standard unit anyway). Regardless of issues with interpretability (which can be noted in the legend), the reviewers felt strongly that plotting these values was a basic matter of convention related to providing a scale and baseline (e.g., 0).

We have created a new version of the figure, including both the axes values, and the outliers (clearly marked).

4) The correlation between SRS and target ROI connectivity at pre-training is potentially problematic, as these behavioral and connectivity scores provide the baseline for the difference metrics used to assess the relationship of brain-behavior changes (Figure 7). How does baseline behavior/connectivity contribute to this change relationship? What if you partial out or control for the baseline?

We thank the reviewers for raising this point. We have carried out this analysis, partialling out both baseline SRS values, and baseline resting state connectivity values. This resulted in numerically stronger, similarly significant correlations between connectivity change and behavior (subsection “Behavioural relevance”).

5) The SRS could be biased by demand characteristics on the part of the parents. How can the authors rule out the possibility of contamination from, for instance, the parents' expectations of positive behavioral change due to neurofeedback training? This limitation should be acknowledged and more details should be reported on the mean SRS before and after training, and the reliability of the difference.

Parents were unaware of the nature of the study and did not perceive it as an intervention, and therefore did not have expectations for behavioral change. In our expanded discussion of the behavioral changes (subsection “Behavioural relevance” and Discussion) we now note that there was no overall change in the mean SRS scores before and after training, which is a further argument against a placebo effect. Note also that a placebo effect cannot explain the correlation between change in scores and change in connectivity, but rather would manifest as an overall shift in behavioral scores.

6) The exclusion of outliers in the SRS change score based on test-retest reliability is problematic. A large change could reflect neurofeedback-based learning that did not (and could not) contribute to basic test-retest reliability in other data. The relationships should be reported with and without the outliers, with a more circumspect discussion about the justification for the exclusions.

We thank the reviewers for pointing out the ambiguity that was present in the original discussion of the SRS results. As is now better clarified in the text (subsection “Behavioural relevance”) and in the figure itself, the outliers showed large negative behavioral change, whereas the prediction for training induced change would be for a positive change. Rather, these were probably an artifact of asking the parents to assess behavior on a timescale of just two weeks, which would overstress any negative incidents which occurred during that time period, creating outliers. We happen to know that one of the participants with outlier data was undergoing a stressful period in his life as his parents volunteered the information that they had just moved to a new house in the week before the post-training questionnaire, but we do not know if there was anything unusual for the other participant, as this was not a routine part of the questionnaire. We would point out though, that after correcting for baseline SRS and rest scores, the correlation had the correct trend even with the outliers (p=0.09).